# Integrated single-cell analysis unveils diverging immune features of COVID-19, influenza, and other community-acquired pneumonia

**Alex R Schuurman[1,2‡]\*, Tom DY Reijnders[1,2‡], Anno Saris[1,2], Ivan Ramirez Moral[1,2], Michiel Schinkel[1,2], Justin de Brabander[1,2], Christine van Linge[1,2], Louis Vermeulen[3], Brendon P Scicluna[1,4], W Joost Wiersinga[1,2,5], Felipe A Vieira Braga[3†], Tom van der Poll[1,2,5†]\***

[1]Center for Experimental and Molecular Medicine, Amsterdam UMC, Academic Medical Center, University of Amsterdam, Amsterdam, Netherlands; [2]Amsterdam Institute for Infection and Immunity, Amsterdam UMC, Amsterdam, Netherlands; [3]Laboratory for Experimental Oncology and Radiobiology, Center for Experimental and Molecular Medicine, Cancer Center Amsterdam and Amsterdam Gastroenterology and Metabolism, Amsterdam UMC, Academic Medical Center, University of Amsterdam, Amsterdam, Netherlands; [4]Division of Infectious Diseases, Amsterdam UMC, Academic Medical Center, University of Amsterdam, Amsterdam, Netherlands; [5]Department of Clinical Epidemiology, Biostatistics and Bioinformatics, Amsterdam UMC, Academic Medical Center, University of Amsterdam, Amsterdam, Netherlands

\*For correspondence:
a.r.schuurman@amsterdamumc.nl (ARS);
t.vanderpoll@amsterdamumc.nl (TP)

†These authors also contributed equally to this work
‡These authors also contributed equally to this work

**Competing interests:** The authors declare that no competing interests exist.

**Abstract** The exact immunopathophysiology of community-acquired pneumonia (CAP) caused by SARS-CoV-2 (COVID-19) remains clouded by a general lack of relevant disease controls. The scarcity of single-cell investigations in the broader population of patients with CAP renders it difficult to distinguish immune features unique to COVID-19 from the common characteristics of a dysregulated host response to pneumonia. We performed integrated single-cell transcriptomic and proteomic analyses in peripheral blood mononuclear cells from a matched cohort of eight patients with COVID-19, eight patients with CAP caused by Influenza A or other pathogens, and four non-infectious control subjects. Using this balanced, multi-omics approach, we describe shared and diverging transcriptional and phenotypic patterns—including increased levels of type I interferon-stimulated natural killer cells in COVID-19, cytotoxic CD8 T EMRA cells in both COVID-19 and influenza, and distinctive monocyte compositions between all groups—and thereby expand our understanding of the peripheral immune response in different etiologies of pneumonia.

## Introduction

The devastating impact of COVID-19 on both a global and interpersonal level is unprecedented in living memory. Patients with severe COVID-19—caused by the SARS-CoV-2 virus—develop bilateral pneumonia, profound systemic inflammation, and organ failure reminiscent of sepsis (*Giamarellos-Bourboulis et al., 2020*; *Huang et al., 2020*; *Richardson et al., 2020*). Within 1 year of its discovery, the world has seen hundreds of thousands excess deaths (*Weinberger et al., 2020*).

Progression to life-threatening disease is mediated by a dysfunctional and excessive host immune response to SARS-CoV-2 (*Wiersinga et al., 2020*). Distinct patterns have emerged in the

characteristics of this immune response in hospitalized patients: lymphopenia (*Huang et al., 2020*; *Zheng et al., 2019*), a role for activated and exhausted T cells (*Chen and John Wherry, 2020*; *De Biasi et al., 2020*; *Stephenson et al., 2021*), hyperinflammatory monocytes with reduced antigen-presenting capacity (*Merad and Martin, 2020*; *Ren et al., 2021*; *Schulte-Schrepping et al., 2020*), delayed or dysfunctional interferon responses (*Lee et al., 2020*; *Wilk et al., 2020*; *Zhang et al., 2020*), and expansion of plasmablasts and suppressive immature neutrophils in severe disease (*Kuri-Cervantes et al., 2020a*; *Schulte-Schrepping et al., 2020*; *Stephenson et al., 2021*).

Crucially, despite early observations that many of the immune disturbances, clinical symptoms, and organ dysfunctions in COVID-19 can also occur in other infections (*Giamarellos-Bourboulis et al., 2020*; *van der Poll et al., 2017*), most studies lack disease controls. The direct comparison of COVID-19 with another, well-matched infectious disease state is urgently needed to distinguish truly specific immune features from the common characteristics of a dysregulated host response to infection. However, while several studies reported on the immunological response during COVID-19 pneumonia at the single-cell level, such investigations are currently scarce for the broader population of community-acquired pneumonia (CAP).

Here, we performed CITE-Seq—Cellular Indexing of Transcriptomes and Epitopes by Sequencing (*Stoeckius et al., 2017*)—on peripheral blood mononuclear cells (PBMCs) from age-, sex-, and disease severity-matched patients with CAP caused by either SARS-CoV-2, Influenza A, or other pathogens admitted to a non-intensive care ward. By integrating single-cell RNA-sequencing with highly multiplexed surface protein marker detection—akin to classical flow cytometry—we create a high-resolution snapshot of cellular phenotypes and functional states, offering insight into the peripheral immune features of different etiologies of pneumonia.

## Materials and methods

### Subjects and sample collection

All individual subject data such as age, sex, and comorbidities can be found in *Supplementary file 1*. This study was part of the ELDER-BIOME project (clinicaltrials.gov identifier NCT02928367) approved by the medical ethical committee of the Amsterdam UMC—location AMC. Written informed consent was obtained from all participants or their legal representatives. In the context of the ELDER-BIOME project, trained research physicians screened patients older than 18 years admitted between October 2018 and June 2020 to the Amsterdam UMC, Flevohospital, or BovenIJ hospital in the Netherlands. Patients were included if they were admitted to the ward and met all of the following criteria: clinical suspicion of an acute infection of the respiratory tract, defined as the presence of at least one respiratory symptom (new cough or sputum production, chest pain, dyspnea, tachypnea, abnormal lung examination, or respiratory failure) and one systemic symptom (documented fever or hypothermia, leukocytosis or leukopenia), and an evident new or progressive infiltrate, consolidation or pleural effusion on chest X-ray or computed tomography scan. Patients were excluded if there was a clinical suspicion of aspiration pneumonia or hospital-associated pneumonia, or if CAP was not the main reason for admission. Patients were excluded from this study if they were severely immunocompromised by either disease or medication. All COVID-19 patients had reverse transcription (RT-PCR)-confirmed SARS-CoV-2 infection, in combination with a CORADS CT-score of 5 (*De Smet et al., 2021*). Heparin anticoagulated blood was obtained within 48 hr of hospital admission. Age- and sex-matched subjects without an infection were included as controls.

### PBMC isolation and storage

For all patients and controls, heparin anticoagulated whole-blood was processed within 4 hr of sampling. PBMCs were separated by density gradient centrifugation using Ficoll-Paque Plus medium (GE Healthcare Life sciences, Little Chalfont, UK) and washed twice, first with cold phosphate-buffered saline (PBS) and then with cold PBS supplemented with 0.5% sterile endotoxin-free bovine serum albumin (BSA) (Divbio Science Europe, Breda, the Netherlands). PBMCs were resuspended in PBS containing 0.5% BSA and 2 mM EDTA and the number of PBMCs was determined using a Coulter Counter (Beckman Coulter, Woerden, the Netherlands). PBMCs were resuspended in Iscove's modified Dulbecco's medium containing 20% filter-sterilized fetal calf serum and pen/strep, after which an equal part of the same medium containing 20% dimethyl sulfoxide was slowly added while

continuously stirring and working on ice. 3–5 million PBMCs were viably stored in 1.8 ml cryogenic vials (Corning #430388), which were slowly brought to −80°C. After 24–72 hr, the cells were transferred to liquid nitrogen storage until further analysis.

## Sorting and staining

After PBMCs were thawed, the Fixable Viability Dye Kit (eBioscience, San Diego, CA) was used to assess cell viability by FACS analysis (>90% in all samples). 500,000 viable singlet-events were sorted per sample using a Sony SH800 Cell Sorter (Sony Biotechnology, San Jose, CA). The sorted cells were incubated with Fc blocker (CD16/CD32, eBioscience) for 10 min, after which each sample was incubated with TotalSeq Hashtag antibody tags to enable multiplexing and subsequent deconvoluting. After Hashtag staining, the cells were pooled into four pools of five samples, each sample contributing equally to their respective pool. The cells were counted manually by light microscopy and Neubauer chambers. Each pooled sample was then incubated for 30 min with our TotalSeq oligoconjugated antibody panel for later surface protein marker quantification. An overview of all antibodies that were used in the study is shown in *Supplementary file 2*.

## Single-cell library generation

Libraries were generated using the Chromium Single Cell 5′ Library and Gel Bead Kit v1.1 (10x Genomics, Pleasanton, CA) following the manufacturer's instructions. In short, pooled cells were loaded aiming the capture of 10,000 cells per pool. Libraries were generated according to standard protocol (Chromium Next GEM Single Cell V(D)J Reagent Kits v1.1 rev E). The amplified mRNA and antibody-derived tags were divided by size and sequenced separately.

## Sequencing

Samples were sequenced using a Hiseq4000 150PE mode. Each position from the $10\times$ chip was loaded into 1 HiSeq 4000 lane. All the four antibody-derived libraries were pooled together and sequenced in 1 HiSeq4000 lane.

## mRNA, hashtags, and antibody alignment

All the libraries were aligned using CellRanger 3.1 (10x Genomics). For the hashtags and antibody tags, we inputted their sequences as provided by BioLegend (San Diego, CA) and tag structure as informed on the Cellranger website.

## Cell deconvolution

To deconvolute the cells belonging to each sample we used the R package Seurat (v3) (*Stuart et al., 2019*). The outputs derived from CellRanger were used to create two separate objects (one with the transcriptome alignment and one with the antibody plus hashtags [HTO] alignment). Initial objects were created using the function 'Read10X'. We filtered both objects based on the cell barcode to keep only cells that were identified in both the transcriptome and in the antibody alignments. After this cell filtering, we used the function 'CreateSeuratObject' to create a transcriptome-based Seurat object. The antibody-derived data was filtered to maintain only the hashtag counts; later it was appended as a specific assay using the 'CreateAssayObject' function. Antibody reads were normalized using the CLR method. For cell demultiplexing, we used the function 'HTODemux' with the parameters 'init=9', 'nsamples=10,000', and 'positivie quantile' ranging between 0.999 and 0.9999999 per each sample, in order to maximize the number of singlets detected. Individual single cells were finally filtered based on their assigned 'HTO_classification.global'='Singlet'.

## Antibody quantification and normalization

Antibody data was normalized using the Seurat function 'NormalizeData' with the parameters 'normalization.method'='CLR' and 'margin'='2', to indicate a normalization across cells.

## Quality control

The cells were filtered based on number of features (nfeature_RNA), number of genes (nCount_RNA), and percentage of mitochondrial reads (percent.mt) using Seurat 'subset(object,

subset=nCount_RNA>1000 and nCount_RNA<10,000 and nFeature_RNA>200 and percent.mt<20)'. An overview of the median number of genes per sample is depicted in *Supplementary file 3*.

## Data scaling, normalization, and cell cycle correction

We calculated cell cycle scores as following ('CellCycleScoring(Healthy_object_Cycle), s.features=s. genes, g2m.features=g2m.genes, set.ident=TRUE') using a list of S phase and G2M phase genes preloaded in Seurat. The cells were then scaled and normalized using the function 'SCTransform (object, vars.to.regress=c('nCount_RNA', 'percent.mt', 'S.Score', 'G2M.Score')).

## Clustering and visualization

Principal components for each set of cells as shown in individual figures were identified using the 'RunPCA' function. Cells were then further processed for clustering and visualization using the Seurat functions 'FindNeighbors', 'FindClusters', and 'RunUMAP'. The number of principal components used as input was determined by using the 'ElbowPlot' function and identifying the number of the PCs which explain most of the data variance.

## Differential expression analysis

Differential expression analysis was performed using the functions 'FindAllMarkers' or 'FindMarkers' and the following parameters: 'min.pct=0.25, logfc.threshold=0.25, assay='SCT''.

## List of software used

CellRanger 3.1; R version 3.6.3 (2020-02-29), FlowJo V10.7, the R packages: Seurat v3, corrplot, dplyr, Matrix, cowplot, rstatix, ggplot2; Adobe Illustrator 2021.

## Quantification and statistical analysis

All statistical analyses were performed using R version 3.6.3. Enrichment analysis was done by first generating contingency tables with the cell distributions per cluster. Chi-squares for the contingency tables were calculated ('chisq<-chisq.test(table)', the residuals rounded 'round(chisq$residuals, 4)') and correlation plots generated 'corrplot(chisq$residuals, is.cor=FALSE, col=rev(brewer.pal(n=8, name='RdBu')), tl.cex=1, cl.pos='r', cl.ratio=1, cl.length=10)' with the graphical parameters shown adjusted per cluster. Comparisons with more than two residuals of difference at least were considered biologically relevant. In the box and whisker plots data was represented with a median line and a box indicating the interquartile range, with individual data points shown (*Figures 1h*, *2h*, and 4h). Significance was defined as $p<0.05$. Statistical significance was determined using either the two-sided Wilcoxon rank-sum test (*Figure 1h*) or the two-sided Kruskal-Wallis test with post hoc pairwise Dunn's test with Benjamini-Hochberg (BH) p-value adjustment (*Figures 2h* and 4h). All gene expression analyses were corrected for multiple testing using the BH method, with significance defined as an adjusted $p<0.05$.

# Results

## Patient characteristics and clinical outcomes

We profiled PBMCs from eight patients hospitalized for CAP caused by SARS-CoV-2 (henceforth referred to as COVID-19), eight patients hospitalized for CAP caused by either Influenza A or other pathogens (all sampled before the start of the COVID-19 pandemic), and four non-infectious control subjects visiting the outpatient clinic without signs of acute infection (*Table 1* for summary data, *Supplementary file 1* for individual patient data). We matched the groups for age, sex, and—the patients—for disease severity using the Modified Early Warning Score (*Churpek et al., 2017*; *Subbe et al., 2001*). Samples from patients were obtained within 48 hr of admission to the general hospital ward, none of the patients received systemic corticosteroid therapy prior to sampling. Patients with COVID-19 reported a longer duration of symptoms prior to hospital admission. Lymphocyte counts were reduced in both patient groups but not significantly different, whereas neutrophil counts were significantly higher in patients with non-COVID-19 CAP. All patients classified as COVID-19 had a positive PCR for SARS-CoV-2. In the patients without COVID-19, three had a positive PCR for Influenza A—one of whom also had a positive blood culture for *Streptococcus*

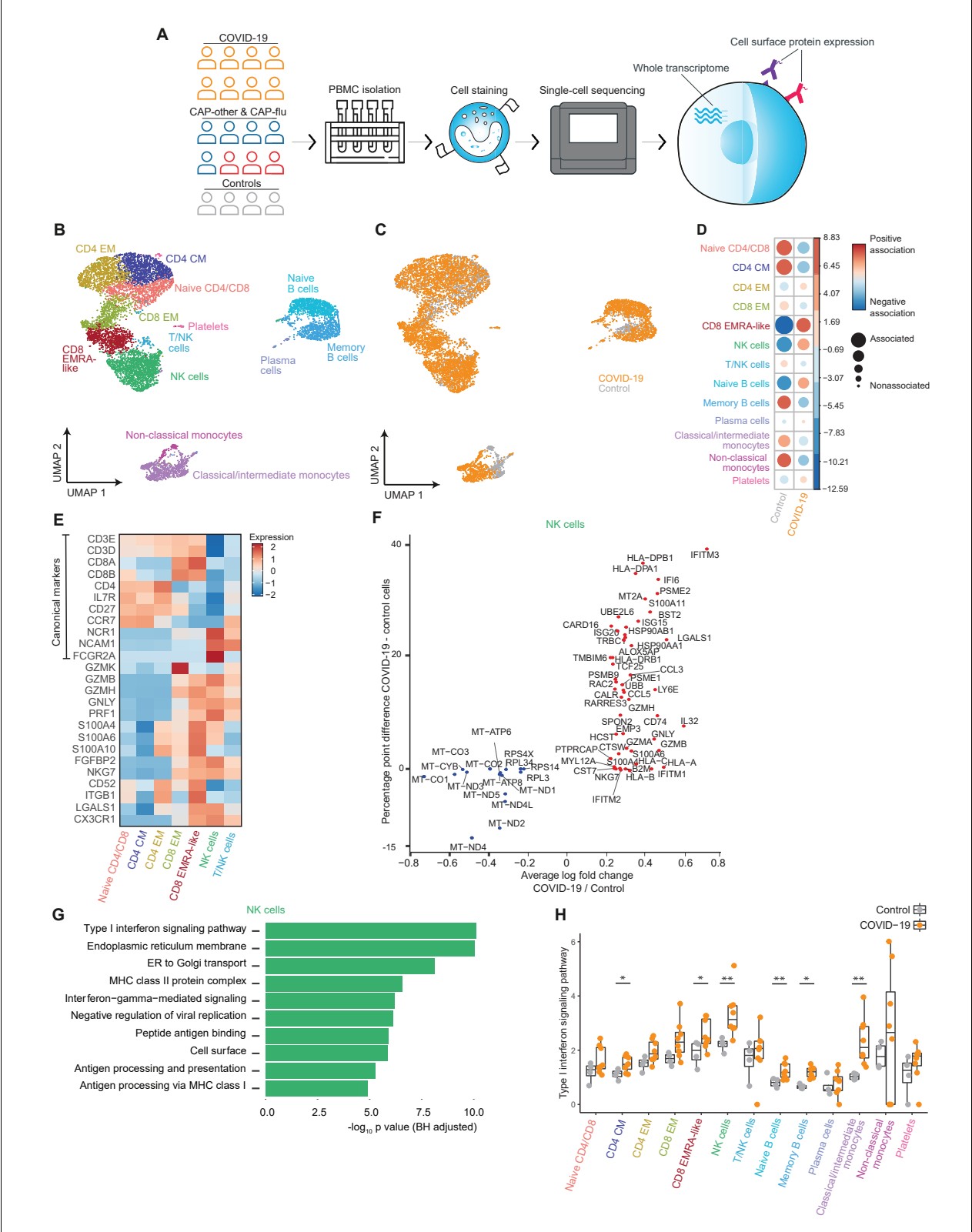

**Figure 1.** The immune response in COVID-19 is characterized by an expansion of CD8 EMRA-like T cells and type I interferon-stimulated NK cells, both demonstrating high cytotoxic potential. (a) Experimental overview: PBMCs from a matched cohort of hospitalized patients with CAP caused by SARS-CoV-2 (COVID-19), CAP caused by Influenza A or other pathogens, and non-infectious controls, were isolated and stained with a panel of oligonucleotide-tagged antibodies. Single-cell mRNA and surface protein expression were subsequently measured on a 10x Genomics platform. (b, c)

*Figure 1 continued on next page*

*Figure 1 continued*

UMAPs depicting the clusters identified by the single-cell transcriptomic analysis of PBMCs from control subjects and patients with COVID-19, each dot representing a single cell. In the first UMAP (**b**), cells are colored by cell type cluster, whereas in the second UMAP (**c**), cells are colored by donor group. See also *Figure 1—figure supplement 1*. (**d**) Correlation plot depicting cluster enrichment in controls and COVID-19 patients. Dot size proportional to Pearson's residual of the chi-squared test (i.e., reflecting the difference between the observed and expected proportion), while the color represents the degree of association from Pearson's chi-squared residuals (red means a positive association, blue means a negative association). (**e**) Heatmap showing the expression of canonical genes and the top differentially expressed genes (DEGs) derived from comparing the CD8 EM and CD8 EMRA-like cell clusters (adjusted p<0.05). The heatmap also shows the expression of these genes in the other identified T and NK cell clusters. See also *Figure 1—figure supplement 2*. (**f**) Graph depicting the DEGs identified when comparing cells from COVID-19 patients and controls within the NK cell cluster. The X-axis depicts the average log fold change and the Y-axis depicts the percentage point difference between the proportion of cells expressing the gene in the COVID-19 group minus the proportion of cells expressing the gene in the control group. All depicted DEGs are statistically significant after adjusting for multiple testing (Benjamini-Hochberg). (**g**) Bar plot showing Gene Ontology pathway analysis of genes upregulated in NK cells from patients with COVID-19 (relative to controls) identified in the analysis in panel (g). The X-axis shows the Benjamini-Hochberg adjusted −log10 p-value from the enrichment score analysis. (**h**) Box and whisker plots showing the enrichment of the type I interferon pathway in all cell subsets, split between COVID-19 patients and controls. The Y-axis depicts the enrichment score. Statistical significance was determined using the two-sided Wilcoxon rank-sum test: *p<0.05, **p<0.01. CAP, community-acquired pneumonia; NK, natural killer; PBMC, peripheral blood mononuclear cell; UMAP, Uniform Manifold Approximation and Projection.

The online version of this article includes the following figure supplement(s) for figure 1:

**Figure supplement 1.** mRNA, surface protein expression, and cell cluster distribution of patients with COVID-19 and non-infectious controls.

**Figure supplement 2.** Comprehensive gene expression profile of T and NK cells in patients with COVID-19 compared with non-infectious controls.

---

*pneumoniae*—and one patient had a positive sputum culture for *Haemophilus influenzae*. In subsequent analyses, we refer to the three patients infected with Influenza A as CAP-flu, and the remaining five patients as CAP-other. None of the patients were admitted to the intensive care unit (ICU) during their hospital stay, one patient with COVID-19 died in the hospital as a result of the disease.

## Highly cytotoxic CD8 EMRA-like T cells and type I interferon-stimulated NK cells characterize COVID-19 patients

An overview of the experimental setup is depicted in *Figure 1a*. Post quality control, we analyzed 16,192 cells from all profiled samples. Throughout this study, we infer disease-specific effects in part by examining the transcriptional states of cell clusters that are proportionally expanded within a disease group. We evaluated proportional differences in cell clusters between groups with the Pearson's residual of the chi-squared test, in which comparisons with more than two residuals of difference were considered biologically relevant. All gene expression analyses were corrected for multiple testing using the BH method, with significance defined throughout as an adjusted p<0.05.

We first explored the peripheral blood immune response of patients with COVID-19 pneumonia and non-infectious controls. Uniform Manifold Approximation and Projection (UMAP) dimensionality reduction of the transcriptome of all individual cells from these subjects identified 13 clusters of myeloid and lymphoid immune cells (*Figure 1b,c*). The top differentially expressed genes (DEGs) between these clusters can be found in *Figure 1—figure supplement 1a*. Cell surface protein expression levels of classical lineage-defining markers (e.g., CD3, CD4, CD8, CD14, and CD19) were consistent with the transcriptome-based clusters (*Figure 1—figure supplement 1b and c*).

When compared with age- and sex-matched non-infectious control subjects, patients with COVID-19 exhibited stark differences in the proportional composition of cell clusters, including a decrease in monocytes and memory B cells, and an increase in naive B cells (*Figure 1c,d*). While some lymphocyte clusters were relatively decreased, we found a striking proportional increase in a specific T cell cluster and NK cells in patients with COVID-19. This expanded T cell cluster mainly expressed low levels of surface CD27 and variable levels of CD45RA (*Figure 1—figure supplement 1d*), resembling the terminally differentiated effector memory re-expressing CD45RA (EMRA) phenotype as defined by classical flow cytometry (CCR7⁻CD27⁻CD45RA⁺; *Martin and Badovinac, 2018*). In fact, these cells (that we will henceforth refer to as CD8 EMRA-like T cells) formed a cluster almost entirely composed of cells from patients with COVID-19 (*Figure 1c* and *Figure 1—figure supplement 1e*). As the CD8 EMRA-like T cell and NK cell clusters showed the most prominent

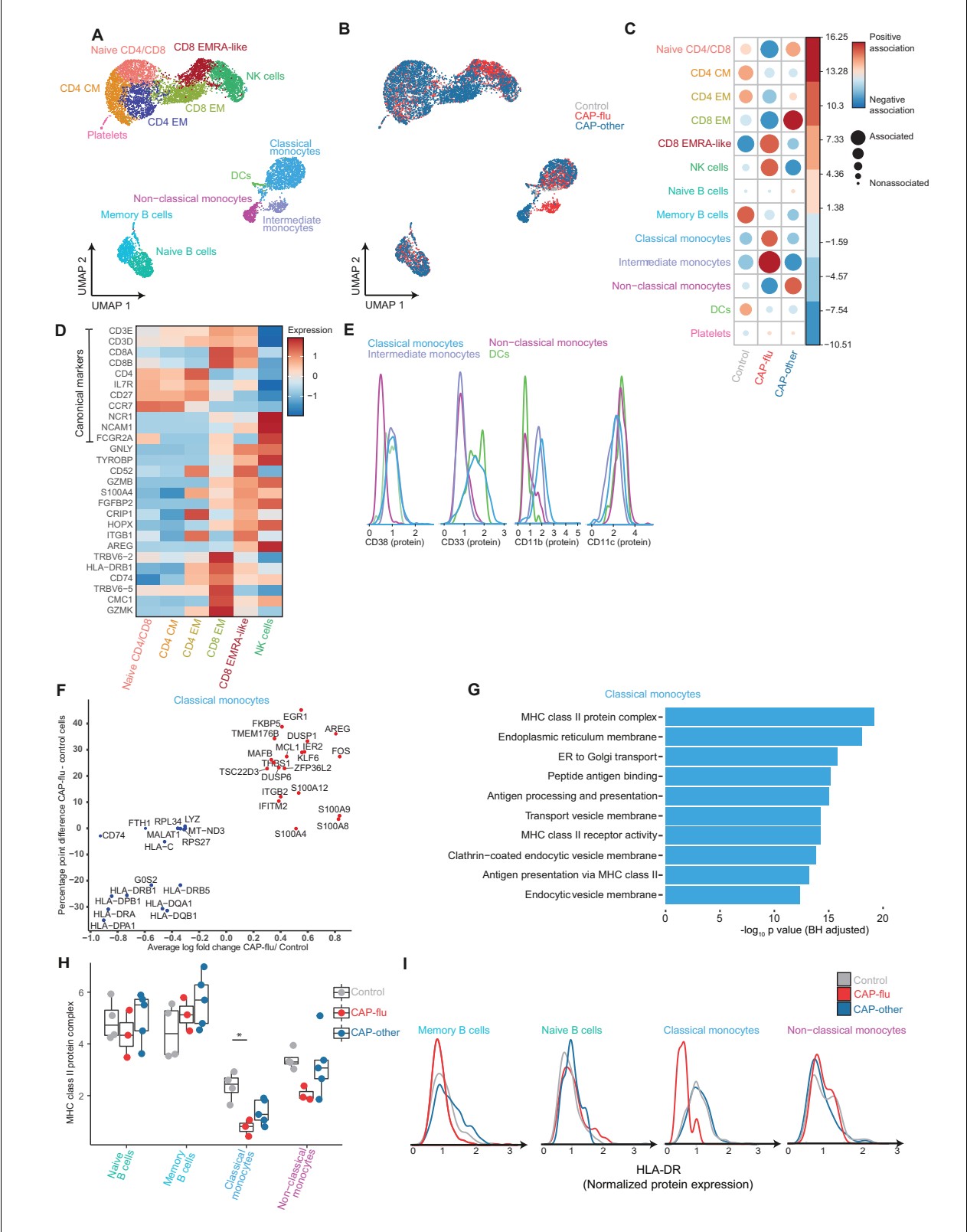

**Figure 2.** The peripheral immune features of T cells, NK cells, and monocytes vary between CAP-flu and CAP-other. (**a, b**) UMAPs depicting the clusters identified by the single-cell transcriptomic analysis of PBMCs from controls, CAP-flu, and CAP-other patients, where each dot represents a single cell. In the first UMAP (**a**), cells are colored by cell type cluster, whereas in the second UMAP (**b**), cells are colored by donor group. See also *Figure 2—figure supplement 1*. (**c**) Correlation plot depicting cluster enrichment in controls, CAP-flu, and CAP-other patients. Dot size proportional to Pearson's

*Figure 2 continued on next page*

*Figure 2 continued*

residual of the chi-squared test (i.e., reflecting the difference between the observed and expected proportion), while the color represents the degree of association from Pearson's chi-squared residuals (red means a positive association, blue means a negative association). (c) Heatmap showing the expression of canonical genes and the top differentially expressed genes (DEGs) derived from comparing the CD8 EM and CD8 EMRA-like cell clusters (adjusted p<0.05). The heatmap also shows the expression of these genes in the other identified T and NK cell clusters. See also *Figure 2—figure supplement 2*. (e) Density plots showing the surface protein expression of CD38, CD33, CD11b, and CD11c per myeloid cell cluster. (f) Graph depicting the DEGs in the classical monocyte cluster when comparing CAP-flu patients versus controls. The X-axis depicts the average log fold change and the Y-axis depicts the percentage point difference between the proportion of cells expressing the gene in the CAP-flu group minus the proportion of cells expressing the gene in the control group. All depicted DEGs are statistically significant after adjusting for multiple testing (Benjamini-Hochberg [BH]). (g) Bar plot showing Gene Ontology pathway analysis of downregulated genes identified in the analysis in panel (g). The X-axis shows the BH adjusted −log10 p-value from the enrichment score analysis. (h) Boxplots depicting the downregulation of the MHC class II protein complex transcriptional pathway in naive B cells, memory B cells, classical monocytes, and non-classical monocytes clusters, split between controls, CAP-flu, and CAP-other patients. Statistical significance was determined using the two-sided Kruskal-Wallis test with post hoc pairwise Dunn's test: *BH-adjusted p<0.05. (i) Density plot showing the normalized surface protein expression of HLA-DR on cells in naive B cells, memory B cells, classical monocytes, and non-classical monocytes clusters, split between controls, CAP-flu, and CAP-other patients. CAP, community-acquired pneumonia; NK, natural killer; PBMC, peripheral blood mononuclear cell; UMAP, Uniform Manifold Approximation and Projection.

The online version of this article includes the following figure supplement(s) for figure 2:

**Figure supplement 1.** mRNA, surface protein expression, and cell cluster distribution of controls, patients with CAP-flu, and patients with CAP-other.

**Figure supplement 2.** Differentially expressed genes of T cell, NK cell, and monocyte clusters between patients with CAP-flu, CAP-other, and control subjects.

---

proportional increase in COVID-19 when compared with non-infectious controls (*Figure 1D*), we next sought to uncover the identity and transcriptional state of these clusters.

In all T and NK cell clusters, we examined the expression of canonical genes and the top DEGs derived from comparing the CD8 EMRA-like and CD8 EM (effector memory) clusters (*Figure 1e* for selected genes, *Figure 1—figure supplement 2a* for the full list of DEGs). At the transcript level, the EMRA-like T cells expressed low levels of the canonical markers *IL7R* (encoding for CD127) and *CCR7*. The CD8 EMRA-like T cells exhibited remarkably high expression of genes related to cytotoxicity and killing—seemingly on par with NK cells—such as *GZMB, GZMH, PRF1,* and *NKG7*. These cells were also marked by the high expression of *ITGB1* (CD29), which has recently been described as a marker of T cells with increased cytotoxicity (*Nicolet et al., 2020*).

We next examined the top DEGs between the CD8 EMRA-like T cell and NK cell clusters. The NK cell cluster was enriched in genes corresponding to type I and, to a lesser extent, type II interferon signaling pathways (*Figure 1—figure supplement 2b,c*). We then compared the expression levels of genes within the NK cell cluster between patients with COVID-19 and control subjects (*Figure 1f*) and discovered marked upregulation of pathways related to a type I interferon response and other antiviral responses in patients with COVID-19 (Biological Process Gene Ontology pathway analysis; *Figure 1g*). Transposing this NK cell cluster-derived type I interferon response signature to the other cell clusters revealed a broad pattern of type I interferon responses across all clusters, statistically significant in several clusters, including NK cells (two-sided Wilcoxon rank-sum test p<0.05; *Figure 1h*). Taken together, when compared with matched non-infectious controls, patients with COVID-19 exhibited a marked proportional increase in CD8 EMRA-like T cells and type I interferon-stimulated NK cells, both with high cytotoxic potential.

## Differential composition of lymphoid cells in patients with CAP-flu and patients with CAP-other

We next investigated the immune features of patients with CAP caused by Influenza A (CAP-flu) or other pathogens (CAP-other), as compared with matched non-infectious controls. UMAP dimensionality reduction revealed 13 clusters of cells (*Figure 2a,c*; and *Figure 2—figure supplement 1a–c* for top DEGs and surface protein expression). Akin to what we observed in patients with COVID-19, patients with CAP-flu showed a proportional increase of CD8 EMRA-like T cell and NK cell clusters (*Figure 2b,c*; cluster distribution per subject in *Figure 2—figure supplement 1d*). Patients with CAP-other had higher levels of the CD8 EM and naïve T cell clusters (*Figure 2c*). Both CAP-flu and CAP-other patients showed a decrease in memory B cells when compared with control subjects (*Figure 2c*).

**Table 1.** Clinical characteristics and disease course.

ACE = angiotensin-converting enzyme; AT-II = angiotensin II; CAP = community-acquired pneumonia; CURB-65 = confusion, blood urea nitrogen, respiratory rate, blood pressure, age 65 or older; COPD = chronic obstructive pulmonary disease; qSOFA = quick sequential organ failure assessment score.

| | COVID-19 (n=8) | CAP* (n=8) | Controls (n=4) |
|---|---|---|---|
| Demographics | | | |
| Age (years) | 66.9 (9.4) | 70.9 (14.3) | 72.2 (1.7) |
| Sex (male) | 5 | 5 | 2 |
| Body mass index | 32.8 (6.5) | 23.8 (7.6) | 25.7 (4.9) |
| Race (white/black) | 4/4 | 7/1 | 4/0 |
| Chronic comorbidities | | | |
| COPD | 0 | 3 | 0 |
| Asthma | 0 | 2 | 0 |
| Hypertension | 4 | 3 | 1 |
| History of myocardial infarction | 0 | 2 | 0 |
| History of stroke | 1 | 0 | 1 |
| Diabetes mellitus, type 2 | 3 | 1 | 1 |
| Chronic kidney disease | 0 | 1 | 1 |
| Chronic medications | | | |
| Inhaled corticosteroids | 0 | 2 | 0 |
| Low-dose oral corticosteroids[†] | 0 | 1 | 0 |
| ACE-inhibitor/AT-II antagonist | 4 | 3 | 1 |
| Statins | 3 | 1 | 3 |
| Platelet aggregation inhibitors | 2 | 1 | 2 |
| Laboratory tests | | | |
| Platelets ($\times 10^9$/L) | 284 (112) | 294 (85) | |
| Leukocytes ($\times 10^9$/L) | 5.7 [2.6, 7.2] | 13.9 [6.0, 19.5] | |
| Neutrophils ($\times 10^9$/L) | 4.4 [1.5, 5.8] | 11.7 [5.3, 17.9] | |
| Lymphocytes ($\times 10^9$/L) | 0.8 [0.4, 1.5] | 1.1 [0.5, 3.1] | |
| Severity scores[‡] | | | |
| Modified Early Warning Score | 3.5 [1.0, 5.0] | 3.5 [2.0, 6.0] | |
| Pneumonia Severity Index | 3.0 [2.0, 4.0] | 3.5 [1.0, 5.0] | |
| CURB-65 | 1.0 [0.0, 2.0] | 1.0 [0.0, 3.0] | |
| qSOFA | 1.0 [0.0, 1.0] | 1.0 [0.0, 1.0] | |
| Disease course | | | |
| Symptoms to admission (days) | 10.0 [2.0, 14.0] | 3.5 [2.0, 9.0] | |
| Hospital length of stay (days) | 3.5 [1.0, 6.0] | 3.0 [2.0, 8.0] | |
| 28 day mortality | 1 | 0 | |

Continuous data are presented as mean (standard deviation) or median (range). Categorical data are presented as counts.

* Caused by either Influenza A, bacterial, or unknown pathogens.

† Corticosteroids<7.5mg prednisolone/day.

‡ Measured upon presentation to the emergency department.

To elucidate the transcriptional states of the differentially expanded T cell subsets between CAP-flu and CAP-other patients, we examined the top DEGs between CD8 EM T and CD8 EMRA-like cells (*Figure 2d*): among the top upregulated genes in the CD8 EM-cluster, we identified *GZMK* and genes related to MHC class II (*HLD-DRB1* and *CD74*). CD8 EMRA-like T cells were characterized by genes related to cytotoxicity and activation signals, such as *GNLY*, *GZMB*, *TYROBP*, and *FGFBP*, which were also highly expressed by NK cells. We observed few specific gene differences in EMRA-like T cells and NK cells from patients with CAP-flu and CAP-other when compared with controls (*Figure 2—figure supplement 2a–d*). Thus, patients with CAP-flu harbored higher proportions of activated and cytotoxic CD8 EMRA-like T cells and NK cells, while the lymphocyte composition in patients with CAP was characterized by higher proportions of the CD8 EM and naïve T cell clusters.

## Classical monocytes from patients with CAP-flu show concurrent signs of inflammation and immune suppression

We next focused on monocytes, because specific clusters were positively associated with either CAP-flu or CAP-other. Patients with CAP-flu exhibited a clear increase in classical monocytes when compared with patients with CAP-other and control subjects (*Figure 2b,c*). The proportional increase in the intermediate monocyte cluster in CAP-flu is unlikely to represent a disease-specific process, as these cells were almost entirely derived from one patient (*Figure 2—figure supplement 1d*). Patients with CAP-other demonstrated a clear increase in non-classical monocytes, contributing nearly all cells in this cluster (*Figure 2b* and *Figure 2—figure supplement 1d*). Low absolute dendritic cell (DC) counts precluded a valid comparison between disease states.

To further explore the myeloid cell clusters that were differentially expanded between CAP-flu and CAP-other, we assessed the surface marker expression of CD14 and CD16 (*Figure 2—figure supplement 2e*). In line with their transcriptome, classical monocytes were mostly CD14$^+$ and CD16$^-$, whereas non-classical monocytes were CD14$^{dim}$ and CD16$^+$. Intermediate monocytes and DCs showed variable CD14 expression, and mostly low CD16 expression (*Figure 2—figure supplement 2e*). Analysis of protein expression identified differential expression patterns of CD11b, CD11c, and CD33, which further consolidated the identity of these clusters (*Figure 2e*; *Sampath et al., 2018*). Differential gene expression analysis directly comparing classical with non-classical monocytes and Gene Ontology pathways corresponding to the upregulated genes are depicted in *Figure 2—figure supplement 2f–h*.

As cell numbers within the classical monocyte cluster were higher among patients with CAP-flu (*Figure 2c*), we examined DEGs within this cluster between CAP-flu and control subjects (*Figure 2f*). Classical monocytes from patients with CAP-flu displayed upregulation of a variety of genes involved in pro-inflammatory processes, such as *EGR-1* (*Pang et al., 2020*), *FKB5* (*Zannas et al., 2019*), and *AREG* (*Zaiss et al., 2015*). Classical monocytes also transcribed several genes encoding for the S100 protein family, such as S100A4, S100A8, S100A9, and S100A12, which were recently implicated in COVID-19 (*Ren et al., 2021*). Strikingly, we observed a concurrent substantially reduced expression of genes related to MHC class II (*Figure 2f*), a quintessential feature of sepsis-induced immune suppression that has been associated with secondary infections and long-term mortality (*Hotchkiss et al., 2013*; *Venet and Monneret, 2018*). Gene Ontology enrichment of these downregulated genes confirmed downregulation of pathways related to antigen presentation (*Figure 2g*) and this pattern was also visible—albeit to a lesser extent and not statistically significant—in non-classical monocytes from patients with CAP-flu and classical monocytes from patients with CAP-other (two-sided Kruskal-Wallis test with post hoc Dunn's test BH-adjusted p<0.05 for CAP-flu vs. control in classical monocytes; *Figure 2h*). Reduced HLA-DR expression on the cell surface of classical monocytes in patients with CAP-flu confirmed this transcriptional pattern at the protein level (*Figure 2i*). Taken together, we report clearly diverging monocyte compositions in patients with CAP-flu and CAP-other, with classical monocytes in CAP-flu expressing pro-inflammatory genes while simultaneously showing immune suppressive features.

## Divergent composition of major immune cell types in patients with COVID-19, CAP-flu, and CAP-other

Next, we directly compared COVID-19, CAP-flu, and CAP-other to delineate the shared and unique immune features between these disease groups. UMAP dimensionality reduction of all cells of these

groups revealed 15 clusters of myeloid and lymphoid cells (the full UMAP of all identified clusters, the distribution per individual patient, and the DEGs and proteins between all clusters are depicted in *Figure 3—figure supplements 1* and *2*). To create an overview of the differences in immune composition between disease states, we first grouped these clusters into metaclusters representing the major cell types: T cells, NK cells, B cells, monocytes, and platelets (*Figure 3a–c*; canonical genes and lineage-defining surface markers in *Figure 3d and e*, respectively). Comparing the proportional composition of immune cells between diseases revealed a significant expansion of NK cells in COVID-19, an expansion of monocytes in CAP-flu, and an overall expansion of T cells in CAP-other (*Figure 3c*). Proportions of B cells were comparable between the groups. As we previously noted reduced HLA-DR surface protein expression in classical monocytes of patients with CAP-flu, we compared the expression of this protein in all monocytes between the three disease states (*Figure 3f*). Overall monocyte HLA-DR surface protein expression in COVID-19 was on par with CAP-flu, and lower than in CAP-other.

## Differences in transcriptional signature between COVID-19 and CAP-flu in T and NK cells

To ascertain differences in the T and NK cell compositions between COVID-19, CAP-flu, and CAP-other, we further examined clusters within these metaclusters (*Figure 4a*; see *Figure 3—figure supplement 1a* for full UMAP of all identified clusters, and *Figure 3—figure supplement 1b* for clusters per individual patient). We observed fewer cells in the naive CD4/CD8 T cell cluster in COVID-19 and CAP-flu as compared with CAP-other, whereas the CD8 EMRA-like 2 T cell cluster was clearly expanded in CAP-flu (*Figure 4b*; see *Figure 3—figure supplement 1c* for the correlation plot of all identified clusters). Patients with CAP-other exhibited a proportional increase of CD8 EM T cells and had the most naive T cells.

To infer functional differences in T cell subsets, we looked at the top DEGs between the two EMRA-like T cell clusters (*Figure 4c*). We noted increased expression of genes related to cytotoxicity, activation and inflammation—such as *GZMB, GNLY, TYROBP, HOPX*, and *CCL5*—in the CD8 EMRA-like two cluster that was expanded in CAP-flu. The contrast in expression levels of these genes was especially apparent in comparison with the CD8 EM T cell cluster predominant in CAP-other, indicating that patients with a known viral infection harbor more activated CD8 T cells.

In line with high *GNLY* expression in the EMRA-like 2 T cell cluster expanded in CAP-flu, we found a higher proportion of granulysin+ NK cells in CAP-flu, whereas the other (granulysin negative) NK cell cluster was highest in COVID-19 (*Figure 4b,c*). Interestingly, the NK cell cluster expanded in COVID-19 expressed higher levels of the interferon-stimulated gene *IFITM3* (*Figure 4d*) than the cluster expanded in CAP-flu. This high expression was even more apparent when comparing cells within the granulysin+ NK cell cluster between patients with COVID-19 and CAP-flu (*Figure 4d*). Gene Ontology enrichment of the genes upregulated in granulysin+ NK cells from patients with COVID-19 exposed a clear type I interferon response (*Figure 4f*). When comparing this pathway between disease groups in all T and NK cell clusters, we found this pronounced type I interferon signature among virtually all clusters in patients with COVID-19, particularly in the EMRA-like T cell clusters and both NK cell clusters (*Figure 4g,h*, statistically significant in the CD8 EMRA-like 2 and granulysin+ NK cell clusters, two-sided Kruskal-Wallis test with post hoc Dunn's test BH-adjusted $p < 0.05$). To summarize, while both groups of patients with known viral infections exhibited high proportions of circulating NK cells and activated (EMRA-like) CD8 T cells when compared with CAP-other, COVID-19 was distinguished from other etiologies of CAP by a pronounced type I interferon signaling transcriptional signature.

## Distinctive monocyte subsets from patients with COVID-19, CAP-flu, and CAP-other

Next, we focused on the differences in monocyte clusters between COVID-19, CAP-flu, and CAP-other (*Figure 5a*; *Figure 3—figure supplement 1a–c*). Classical monocytes were proportionally expanded in both CAP groups, albeit primarily in patients with CAP-flu (*Figure 5b*). The intermediate monocyte cluster was prominently expanded in patients with COVID-19, whereas the non-classical monocyte cluster showed a clear expansion in the CAP-other patients.

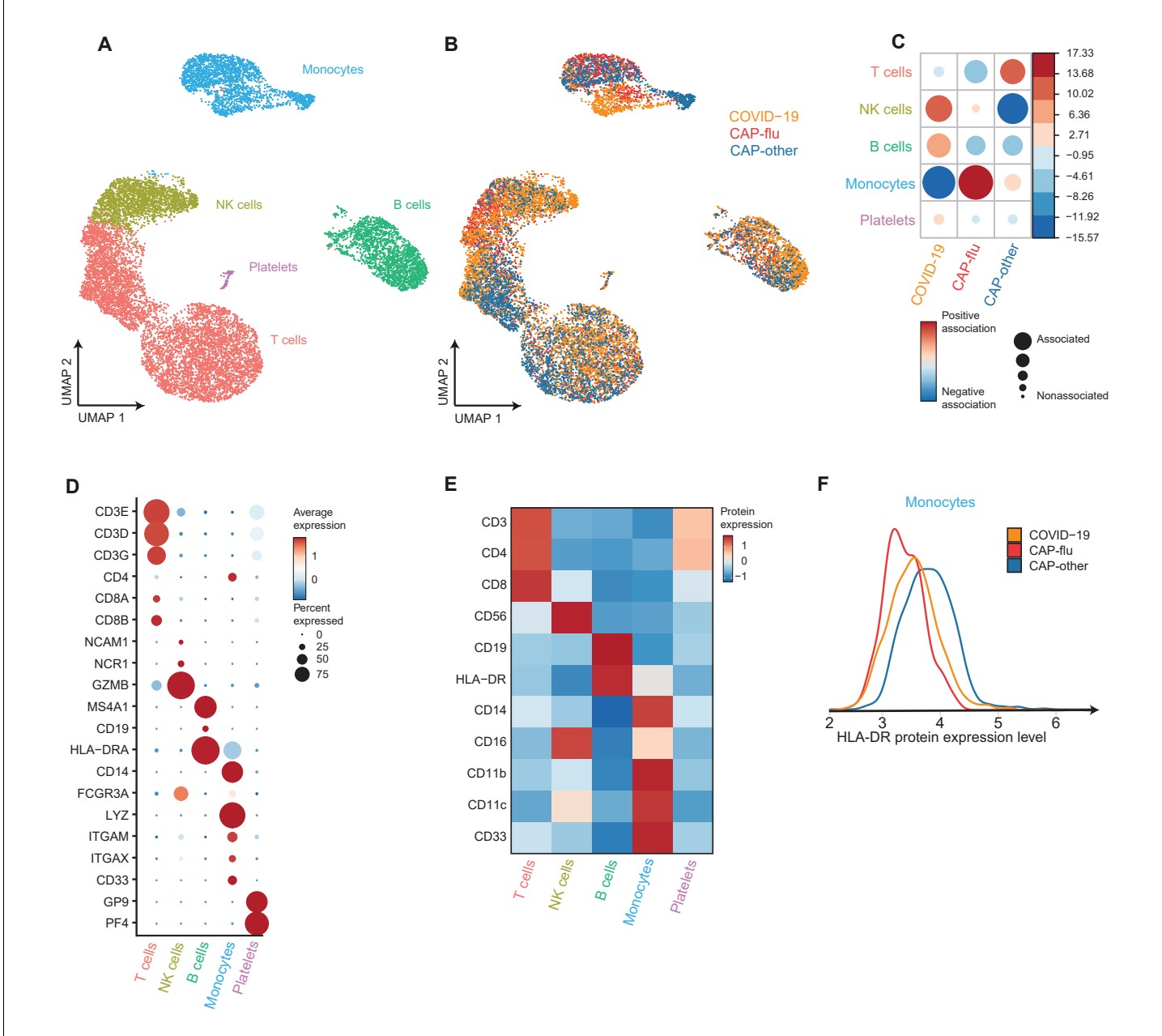

**Figure 3.** Divergent composition of major immune cell types in patients with COVID-19, CAP-flu, and CAP-other. (**a, b**) UMAPs depicting the metaclusters identified by the single-cell transcriptomic analysis of PBMCs from controls, COVID-19, CAP-flu, and CAP-other patients, where each dot represents a single cell. In the first UMAP (**a**), cells are colored by cell type cluster, whereas in the second UMAP (**b**), cells are colored by donor group. (**c**) Correlation plot depicting metacluster enrichment in COVID-19, CAP-flu, or CAP-other patients. Dot size proportional to Pearson's residual of the chi-squared test (i.e., reflecting the difference between the observed and expected proportion), while the color represents the degree of association from Pearson's chi-squared residuals (red means a positive association, blue means a negative association). (**d**) Dot plot showing canonical genes per identified metacluster. Color indicates the normalized level of expression, while the dot size is proportional to the percentage of cells per cluster expressing the canonical gene. (**e**) Heatmap showing the expression of lineage-defining protein surface markers per metacluster. (**f**) Density plot showing the normalized surface protein expression of HLA-DR on the monocyte lineage, split between COVID-19, CAP-flu, and CAP-other patients. CAP, community-acquired pneumonia; UMAP, Uniform Manifold Approximation and Projection.

The online version of this article includes the following figure supplement(s) for figure 3:

**Figure supplement 1.** Cell clusters in individual samples from COVID-19, CAP-flu, and CAP-other.

**Figure supplement 2.** Differential mRNA and surface protein expression between COVID-19, CAP-flu, and CAP-other patients.

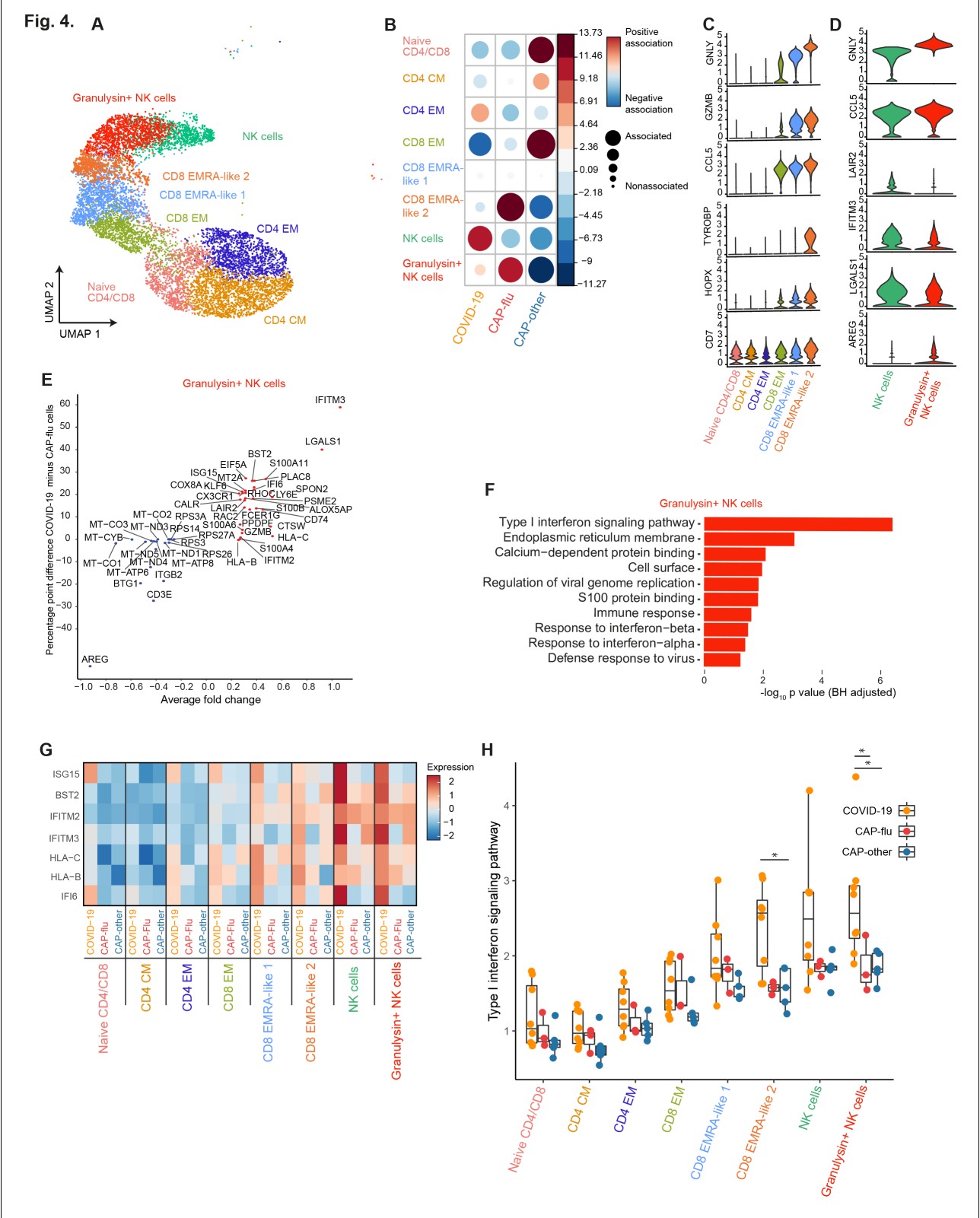

**Figure 4.** CAP-flu is characterized by expansion of T cells and NK cells expressing granulysin, while COVID-19 T and NK cells exhibit a clear type I interferon signature. (a) UMAP depicting the T and NK cell clusters identified by the single-cell transcriptomic analysis of PBMCs from COVID-19, CAP-flu, and CAP-other patients, where each dot represents a single cell with each color corresponding to a specific cell type cluster. The full UMAP of all identified clusters, the distribution per individual patient, and the differentially expressed genes and proteins between all clusters are depicted in

*Figure 4 continued on next page*

*Figure 4 continued*

*Figure 3—figure supplements 1* and *2*. (**b**) Correlation plot depicting T and NK cell cluster enrichment in COVID-19, CAP-flu, and CAP-other. Dot size proportional to Pearson's residual of the chi-squared test (i.e., reflecting the difference between the observed and expected proportion), while the color represents the degree of association from Pearson's chi-squared residuals (red means a positive association, blue means a negative association). See also *Figure 3—figure supplement 1c* for the correlation plot of all identified clusters. (**c**) Violin plots showing the expression of the top DEGs derived from comparing the two identified EMRA-like T cell clusters (adjusted p<0.05). The expression of these genes in other T cell clusters is also depicted. See also *Figure 3—figure supplement 2* for the top DEGs and surface protein expression differences between all identified cell clusters. (**d**) Violin plots showing the expression of the top DEGs between the two identified NK cell clusters (adjusted p<0.05). (**e**) Graph depicting the DEGs identified when comparing COVID-19 cells and CAP-flu cells in the granulysin+ NK cell cluster. The X-axis depicts the average log fold change and the Y-axis depicts the percentage point difference between the proportion of cells expressing the gene in the COVID-19 group minus the proportion of cells expressing the gene in the CAP-flu group. All depicted DEGs are statistically significant after adjusting for multiple testing (Benjamini-Hochberg [BH]). (**f**) Bar plot showing Gene Ontology pathway analysis of the genes upregulated in granulysin+ NK cells from patients with COVID-19. The X-axis shows the BH adjusted −log10 p-value from the enrichment score analysis. (**g**) Heatmap showing the expression of genes in the type I interferon signaling pathway in all T and NK cell subsets, split between COVID-19, CAP-flu, and CAP-other patients. (**h**) Box and whisker plots depicting the upregulation of the type I interferon signaling pathway in all T and NK cell subsets, split between COVID-19, CAP-flu, and CAP-other patients. Statistical significance was determined using the two-sided Kruskal-Wallis test with post hoc pairwise Dunn's test: *BH-adjusted p<0.05. CAP, community-acquired pneumonia; DEG, differentially expressed gene; NK, natural killer; UMAP, Uniform Manifold Approximation and Projection.

To explore the transcriptional states of the diverging monocyte compositions in disease states, we looked at the top DEGs between the monocyte clusters, and the pathways related to the upregulated genes within each cluster (*Figure 5c–f*). In classical monocytes (proportionally higher in patients with CAP-flu) the majority of upregulated genes was related to ribosomes and viral transcription (*Figure 5c,d*). The intermediate monocytes (consistently present in COVID-19) were characterized by a pronounced antiviral, interferon-driven response, as illustrated by the upregulation of both interferon-inducible and stimulated genes (*IFI6, IFI27, IFI44L,* and *ISG15, SIGLEC1*; *Figure 5c, e*). The high expression of *APOBEC3A* and *MX1*, involved in restricting viral activity in monocytes through RNA editing (*Sharma et al., 2015*; *Verhelst et al., 2012*), underlined this increased antiviral potential. The upregulated DEGs of the non-classical monocyte cluster (predominantly present in CAP-other) were more heterogeneous: they expressed genes involved in MHC class II antigen-presenting, Fc receptors (*FCGR3A* and *FCER1G*), the complement system (*C1QA* and *CFD*), and multiple genes associated with cell activation (*AIF1, LST1,* and *SMIM25*; *Figure 5c,f*).

An overview of NK cell, T cell, and monocyte differences between COVID-19, CAP-flu, and CAP-other is shown in *Table 2*.

## Discussion

We performed single-cell transcriptomic and proteomic analyses in PBMCs from a matched cohort of non-infectious control subjects, patients with CAP caused by SARS-CoV-2 (COVID-19), Influenza A, or other pathogens. All patient samples were collected within 48 hr of admission to a non-intensive care hospital ward.

Despite overall lymphopenia in our population of patients with COVID-19, consistently described in earlier studies (*Wiersinga et al., 2020*), we report a proportional increase in cells that resemble CD8 EMRA T cells with a pronounced cytotoxic transcriptional signature. Importantly, CAP-flu—but not CAP-other—was associated with a similar CD8 EMRA T cell response, suggesting that this is induced by viral infection in general rather than by SARS-CoV-2 specifically. CD8 EMRA T cells effectively kill virus-infected cells, have a varying proliferation capacity, and (together with CD8 EM T cells) represent the predominant phenotype in circulating CD8 T cells specific for respiratory viruses in adults (*Martin and Badovinac, 2018*; *Schmidt and Varga, 2018*; *Verma et al., 2017*). Other studies in COVID-19 similarly reported CD8 T cells that were highly activated (*De Biasi et al., 2020*; *Kuri-Cervantes et al., 2020b*; *Mathew et al., 2020*; *Zhang et al., 2020*) had increased cytotoxic potential (*Kuri-Cervantes et al., 2020b*; *Zhang et al., 2020*) and exhibited characteristics of EMRA (or sometimes EM) cells (*De Biasi et al., 2020*; *Mathew et al., 2020*; *Neidleman et al., 2020*; *Weiskopf et al., 2020*). These T cells may be clonally expanded in moderate, but not severe, disease (*Liao et al., 2020*; *Zhang et al., 2020*). Taken together, these findings indicate an important role for cytotoxic T cells with EMRA-like features in controlling the virus both in influenza and COVID-19 pneumonia, although it would be informative to contrast these findings with the T cell response in

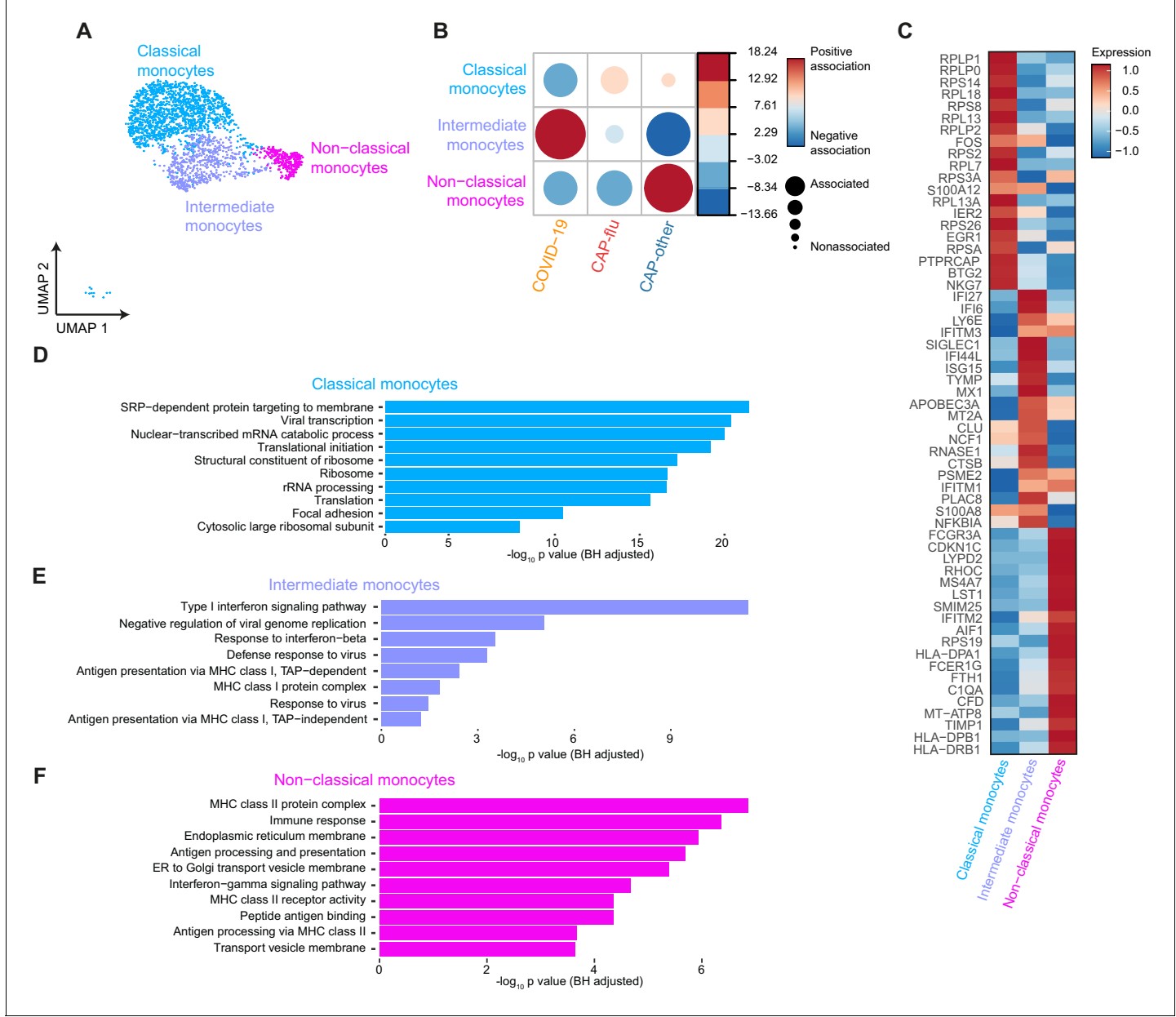

**Figure 5.** Distinctive subset compositions and transcriptional profiles in monocytes from patients with COVID-19, CAP-flu, and CAP-other. (a) UMAP depicting the monocyte cell clusters identified by the single-cell transcriptomic analysis of PBMCs from COVID-19, CAP-flu, and CAP-other patients, where each dot represents a single cell with each color corresponding to a specific cell type cluster. The full UMAP of all identified clusters and the distribution per individual patient is depicted in *Figure 3—figure supplement 1a,b*. (b) Correlation plot depicting cluster enrichment in COVID-19, CAP, and CAP-flu patients. Dot size proportional to Pearson's residual of the chi-squared test (i.e., reflecting the difference between the observed and expected proportion), while the color represents the degree of association from Pearson's chi-squared residuals (red means a positive association, blue means a negative association). (c) Heatmap showing the expression of the top DEGs (adjusted p<0.05) between all three monocyte clusters. (d) Gene Ontology pathway analysis of the upregulated DEGs in classical monocytes, (e) intermediate monocytes, and (f) non-classical monocytes. The X-axis shows the Benjamini-Hochberg adjusted −log10 p-value from the enrichment score analysis. CAP, community-acquired pneumonia; DEG, differentially expressed gene; UMAP, Uniform Manifold Approximation and Projection.

infections that do not require hospitalization. Similarly, it would be valuable to compare these data with patients that have a more complicated outcome, such as ICU admission, to further separate immunopathology from a homeostatic immune response.

**Table 2.** Overview of proportionally increased cell clusters and transcriptional characteristics in the direct comparison between COVID-19, CAP-flu, and CAP-other.

| | COVID-19 | CAP-flu | CAP-other |
|---|---|---|---|
| NK cells | Granulysin+ NK cells ↑ *higher type I interferon response than CAP-flu*<br>NK cells *high IFITM3 expression* | Granulysin+ NK cells ↑↑ *higher AREG expression than COVID-19* | Low numbers of NK cells |
| T cells | CD8 EMRA-like 2 ↑ *cytotoxicity, activation and inflammation* | CD8 EMRA-like 2 ↑↑ *cytotoxicity, activation and inflammation* | Naive CD4/CD8 *not activated*<br>CD8 EM *less activated than EMRA-like 1 and 2 clusters* |
| Monocytes | Intermediate monocytes *high type I interferon response, increased antiviral potential* | Classical monocytes *ribosomal and viral transcription genes* | Non-classical monocytes *heterogenous gene expression* |

Whereas T cell lymphopenia has been a ubiquitous finding, the effect of COVID-19 on NK cell numbers is more ambiguous, with cytopenia often limited to the most severe cases (*Kuri-Cervantes et al., 2020b*; *Laing et al., 2020*; *Wilk et al., 2020*; *Zhang et al., 2020*). We report a proportional increase in highly cytotoxic NK cells with a type I interferon transcriptional signature in COVID-19. Patients with CAP-flu—but not with CAP-other—also displayed an expansion of NK cells when compared with non-infectious control subjects, but this was less pronounced when compared directly with COVID-19. While the cytotoxic and interferon-responsive transcriptional features are in line with earlier findings, an expansion of NK cells has not been described in previous COVID-19 single-cell reports (*Wilk et al., 2020*; *Zhang et al., 2020*; *Zhu et al., 2020*). However, Wilk et al. reported that CD56^dim NK cells—key in the cytotoxic antiviral response—were maintained in moderate disease, whereas the CD56^bright NK cells (prone to produce pro-inflammatory cytokines such as IFN-γ) were depleted in both moderate and severe diseases (*Wilk et al., 2020*). COVID-19 has been hypothesized to either induce hyperresponsive NK cells that exacerbate lung-destructive inflammation through the production of IFN-γ; or to hamper effective viral clearance by NK cells, such as through suppressing type I interferon responses and reducing circulating cell numbers (*Market et al., 2020*). In influenza, depletion of NK cells has been implicated only in severe disease, with normal or even increased NK cell counts after mild disease or vaccination (*Fox et al., 2012*; *Jost et al., 2011*). While our results partly deviate from previous reports, they could indicate a relatively adequate NK cell response in this population of non-ICU patients with COVID-19 and CAP-flu. Overall, the proportional expansion and activation of CD8 T and NK cells, together with a potent interferon response could be indicative of a homeostatic response to a viral infection. While in this regard it would be of interest to compare patients with different disease outcomes, we deemed the number of patients with a complicated outcome in our cohort too small to justify such an analysis.

Monocytes are key players in the innate immune response to pathogens and have been reported to respond divergently to different types of RNA viruses (*de Marcken et al., 2019*). We found striking proportional and transcriptional differences in monocyte subsets between the disease groups: in COVID-19, intermediate monocytes were expanded with a clear antiviral and type I interferon signature, while in CAP-flu, classical monocytes predominated and were characterized by ribosomal translational activity. An enhanced type I interferon response in monocytes of patients with COVID-19 has previously been reported (*Schulte-Schrepping et al., 2020*; *Zhang et al., 2020*), but interestingly CAP-flu did not provoke such a strong antiviral response in the expanded monocyte subset. Interferon evasion and hijacking of the host's ribosomal translational machinery are hallmarks of active viral replication (*Chen et al., 2018*; *García-Sastre, 2011*; *Walsh and Mohr, 2011*). While both Influenza and SARS-CoV-2 can infect human monocytes (*Hoeve et al., 2012*; *Pontelli et al., 2020*; *Ren et al., 2021*), the clear transcriptomic difference in our population could indicate that monocytes in patients with CAP-flu—but not COVID-19—harbored intracellular viruses. Furthermore, we found a profound decrease of HLA-DR—which can be indicative of immune suppression—in classical monocytes both in COVID-19 and CAP-flu, when compared with the CAP-other group. Due to our limited sample size, and as decreased expression of HLA-DR is a hallmark of sepsis in patients with CAP, these relatively low levels of HLA-DR in monocytes from patients with viral CAP should be interpreted with caution (*Venet et al., 2020*). In CAP-other, non-classical monocytes preponderated, characterized by enhanced expression of genes involved in MHC-II signaling. This, together with the

differential lymphocyte responses discussed above, clearly suggests that the peripheral immune response in CAP depends at least in part on the type of causative pathogen.

In the NK cells and (to a lesser extent) non-classical monocytes of patients with COVID-19, we found remarkably high expression of interferon-induced transmembrane protein 3 (*IFITM3*). *IFITM3* plays an essential role in restricting viral replication in CAP-flu (*Yánez et al., 2020*), as mice lacking *IFITM3* developed severe pulmonary inflammation when infected with an influenza virus of limited virulence (*Everitt et al., 2012*). Our results underline a potential role of *IFITM3* in COVID-19, although mechanistic evidence will be decisive.

Strengths of this study include the application of the novel CITE-seq technique, directly integrating proteomic surface marker and transcriptomic data on a single-cell level. This enabled us to readily validate transcriptome-derived clusters and certain disease-related transcriptional patterns at the protein level. Furthermore, the design of this study (in terms of matching and sampling within 48 hr after hospital admission) and the inclusion of disease controls strengthen the observations. Our investigation has several limitations. Experimentally, the total number of analyzed cells was at the lower end of standard practice, which was mainly due to our multiplexed and integrated design. Furthermore, our study had a limited sample size as viable PBMC samples from severity matched, non-COVID-19 CAP patients are scarce. Differences in patient characteristics, such as body mass index and duration of symptoms prior to admission, were largely representative of expected differences between the patient populations (*Wiersinga et al., 2020*). The difference in duration of symptoms prior to hospital admission can be considered a limitation, as time likely influences the composition of immune cell subsets and transcriptional states. We were also unable to take into account the influence of virological features, such as the level of viral load and/or presence of viremia, as these data were not available. The causative organism in CAP is typically only identified in approximately 50% of patients (*Jain et al., 2015*; *Welte et al., 2012*). While in this respect our (non-COVID-19) CAP cohort is representative of a general CAP population, patients with CAP-other likely are a more heterogeneous group than patients with pneumonia caused by one pathogen. Finally, one subject in our cohort was co-infected with both Influenza A and *S. pneumoniae.* We nevertheless decided to include this patient to create a realistic representation of patients with CAP-flu, as up to 30% of laboratory-confirmed pneumonia cases can involve viral and bacterial co-infections (most commonly influenza and *S. pneumoniae*) (*Gupta et al., 2008*; *Jain et al., 2015*).

Collectively, this investigation provides insight into the peripheral immune features of CAP, including COVID-19, at the single-cell level in patients admitted to a general hospital ward. By contrasting multiple commonly encountered forms of CAP, we provide a framework for the roles of T cells, NK cells, and monocytes in the immunopathophysiology of CAP. This knowledge could guide future mechanistic studies seeking pathogen-specific interventions.

## Acknowledgements

This work was supported by the Corona Research Fund from the AMC Foundation. TDYR and AS are supported by NACTAR (# 16447) financed by the Dutch Research Council (NWO). JB and CL are supported by the European Commission (FAIR # 847786).

## Additional information

### Funding

| Funder | Grant reference number | Author |
| --- | --- | --- |
| NWO | 16447 | Tom D Y Reijnders |
| European Commission | 847786 | Justin de Brabander<br>Christine van Linge |
| AMC | Corona Research Fund | Tom D Y Reijnders |

The funders had no role in study design, data collection and interpretation, or the decision to submit the work for publication.

## Author contributions
Alex R Schuurman, Tom DY Reijnders, Conceptualization, Data curation, Formal analysis, Investigation, Visualization, Methodology, Writing - original draft, Writing - review and editing; Anno Saris, Ivan Ramirez Moral, Michiel Schinkel, Investigation, Writing - review and editing; Justin de Brabander, Christine van Linge, Investigation; Louis Vermeulen, Funding acquisition, Writing - review and editing; Brendon P Scicluna, Supervision, Writing - review and editing; W Joost Wiersinga, Resources, Supervision, Funding acquisition, Writing - review and editing; Felipe A Vieira Braga, Data curation, Software, Formal analysis, Visualization, Writing - review and editing; Tom van der Poll, Conceptualization, Resources, Supervision, Funding acquisition, Project administration, Writing - review and editing

## Author ORCIDs
Alex R Schuurman ⬤ https://orcid.org/0000-0001-9322-1117
Tom DY Reijnders ⬤ https://orcid.org/0000-0002-1764-0114
Brendon P Scicluna ⬤ https://orcid.org/0000-0003-2826-0341

## Ethics
Human subjects: This study was part of the ELDER-BIOME project (clinicaltrials.gov identifier NCT02928367) approved by the medical ethical committee of the Amsterdam UMC - location AMC. Written informed consent was obtained from all participants or their legal 643 representatives.

## Decision letter and Author response
Decision letter https://doi.org/10.7554/eLife.69661.sa1
Author response https://doi.org/10.7554/eLife.69661.sa2

# Additional files

## Supplementary files
- Supplementary file 1. Overview of clinical data per individual subject.
- Supplementary file 2. Overview of all antibodies that were used in this study.
- Supplementary file 3. Overview of technical information per sample.
- Transparent reporting form
- Reporting standard 1. Strobe checklist.

## Data availability
Sequencing data have been deposited in GEO under accession code GSE164948.

The following dataset was generated:

| Author(s) | Year | Dataset title | Dataset URL | Database and Identifier |
| --- | --- | --- | --- | --- |
| Schuurman AR, Reijnders TDY | 2021 | Single-cell sequencing data of patients with COVID-19, patients with CAP and controls | http://www.ncbi.nlm.nih.gov/geo/query/acc.cgi?acc=GSE164948 | NCBI Gene Expression Omnibus, GSE164948 |

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
