## [Decision Letter]

**Acceptance summary:**

The authors did a pioneer study comparing single-cell transcriptomic and proteomic signatures from patients with CAP caused by SARS-CoV-2, influenza and other pathogens. While the short number of patients included makes difficult to generalise the results of this study, it has nonetheless a number of major strengths: (1) comparable severity of the three groups; (2) absence of steroidal treatment; (3) the patients are in the first 48 hours following hospitalization at the wards.

**Decision letter after peer review:**

Thank you for submitting your article "Integrated single-cell analysis unveils diverging immune features of COVID-19, influenza and other community-acquired pneumonia" for consideration by *eLife*. Your article has been reviewed by 3 peer reviewers, including Evangelos J Giamarellos-Bourboulis as the Reviewing Editor and Reviewer #2, and the evaluation has been overseen by Jos van der Meer as the Senior Editor. The following individuals involved in review of your submission have agreed to reveal their identity: Jesus Bermejo-Martin (Reviewer #1); Timothy Billiar (Reviewer #3).

Essential revisions:

1. The results support a marked effector T CD8 response along with activated NK and interferon responses in COVID-19 and influenza, which could be expected during an homeostatic response to a viral infection. How do the authors interpret these findings in COVID-19, where a depressed interferon response has been proposed as a signature of severity, at least in some patients? Are the findings here due to the fact that most patients included here resolve the infection? The specific profile of the patient who died could be very informative to this extent.

2. The more patent depression of HLA-DR in viral CAP compared to CAP of other origin here should be interpreted with caution: depressed expression of HLA-DR is a hallmark of sepsis due to CAP.

3. We would appreciate to include a table summarising the common/divergent signatures between the three groups of patients.

4. The authors acknowledge that it would be interesting to compare their results with that obtained in patients with COVID-19 not needing hospitalization. They should also mention that it would be also interesting to compare them with those from a group showing complicated outcomes (ICU admission and death). This limitation makes difficult to elucidate whether the changes described in this work correspond to the process of normal resolution of an infection or otherwise contribute to immunopathology.

5. Another limitation is the absence of virological data to match them with the immunological findings here: duration of viral shedding, viral load in respiratory samples, presence/absence of viremia….This should also be acknowledged.

The authors need to change the way they present the results. The readers are interested to head-to-dead comparisons with the other categories of community-acquired pneumonia (CAP) so Figures 4 and 5 should follow Figure 1.

6. The number of patients are really few; so comparisons should be deleted from Table 1.

7. What were the causative pathogens in non-flu CAP?

8. Were PBMCs isolated from critical COVID-19 or just moderate cases? What can be the impact of severity on the three main clusters? Is this clustering associated with outcome and need for dexamethasone treatment?

9. The discussion is too long. The authors need to condense referral to non-COVID-19 as much as possible.

10. An integrated view across all the groups with an emphasis of the changes not only in cell specific transcriptional responses, but also at the cell-state level would be of interest.

11. The authors are encouraged to provide a complete list of the antibodies used in the CITE seq analysis.

12. Technical information would also be of interest including the cell numbers per sample, the median detected gene numbers (especially in view of the 5' sequencing and access to the full transcriptomic response).

13. A clear limitation is the possibility that some of the major changes between the group reflect the different durations of infection. This may be especially true for the Covid-19 patients where the duration of symptoms was much longer than the other causes of CAP. This should be listed as a limitation of the study.

---

## [Author Response]

Essential revisions:1. The results support a marked effector T CD8 response along with activated NK and interferon responses in COVID-19 and influenza, which could be expected during an homeostatic response to a viral infection. How do the authors interpret these findings in COVID-19, where a depressed interferon response has been proposed as a signature of severity, at least in some patients? Are the findings here due to the fact that most patients included here resolve the infection? The specific profile of the patient who died could be very informative to this extent.

We agree with the reviewer that the proportional expansion and activation of CD8 T and NK cells could be a sign of a homeostatic response to a viral infection. We do indeed think that this may be an indication of a relatively well-functioning immune response.

As for the depressed interferon response to a viral infection, we assume the reviewer is referring to the early paper by Hadjadj and colleagues published in Science, which showed a depressed interferon response in severe (defined as requiring >3L of oxygen) and critically ill patients. Several factors may explain the differences with our study, including methods (bulk RNA-seq vs single-cell) and the fact that our population of ward patients was less severely ill. Furthermore, later studies have also reported heightened interferon response the most severely ill patients. While we share the interest in looking at the interferon response in the patient who died, we think valid inferences cannot be made from the expression profile of a single patient, and therefore prefer not to add this data to the revised manuscript.

To address these points in the manuscript, we added the following sentence to the discussion (page 23-24, line 508-512):

“Overall, the proportional expansion and activation of CD8 T and NK cells, together with a potent interferon response could be indicative of a homeostatic response to a viral infection. While in this regard it would be of interest to compare patients with different disease outcomes, we deemed the number of patients with a complicated outcome in our cohort too small to justify such an analysis.”

2. The more patent depression of HLA-DR in viral CAP compared to CAP of other origin here should be interpreted with caution: depressed expression of HLA-DR is a hallmark of sepsis due to CAP.

We fully agree with the reviewer that we cannot draw firm conclusions about the differences in HLA-DR expression between viral and bacterial infections from these data. Decreased HLA-DR expression is indeed a hallmark of immune suppression in sepsis, although it has also been reported in other critically ill patients, such as those with trauma.

While we have removed our more thorough discussion of this topic to abbreviate the discussion somewhat (in response to comment 9), we have added the following sentence (page 24, line 527- 532) to reflect the reviewer's point:

“Furthermore, we found profound decrease of HLA-DR – which can be indicative of immune suppression – in classical monocytes both in COVID-19 and CAP-flu, when compared with the CAP-other group. Due to our limited sample size, and as decreased expression of HLA-DR is a hallmark of sepsis in patients with CAP, these relatively low levels of HLA-DR in monocytes from patients with viral CAP should be interpreted with caution.”

3. We would appreciate to include a table summarising the common/divergent signatures between the three groups of patients.

Thank you for this suggestion, we added main table which summarizes the main findings for NK cells, T cells and monocytes (Table 2, page 21).

4. The authors acknowledge that it would be interesting to compare their results with that obtained in patients with COVID-19 not needing hospitalization. They should also mention that it would be also interesting to compare them with those from a group showing complicated outcomes (ICU admission and death). This limitation makes difficult to elucidate whether the changes described in this work correspond to the process of normal resolution of an infection or otherwise contribute to immunopathology.

Indeed, it would be very interesting to compare different levels of disease severity, to separate pathology from homeostatic processes. Unfortunately this was not feasible in the current study, we now added this a limitation in the discussion (page 22-23, line 483-486):

“Similarly, it would be valuable to compare these data with patients that have a more complicated outcome, such as ICU admission, to further separate immunopathology from a homeostatic immune response.”

5. Another limitation is the absence of virological data to match them with the immunological findings here: duration of viral shedding, viral load in respiratory samples, presence / absence of viremia….This should also be acknowledged.The authors need to change the way they present the results. The readers are interested to head-to-dead comparisons with the other categories of community-acquired pneumonia (CAP) so Figures 4 and 5 should follow Figure 1.

Thank you for this important remark. We agree that detailed information regarding the viral load would be a valuable addition to this paper, in particular in relation to the described immune parameters. We added the lacking of such data as a limitation in the discussion (page 25-26, line 558-560):

“We were also unable to take into account the influence of virological features, such as the level of viral load and/or presence of viremia, as these data were not available.”

We agree that the comparison between CAP and COVID-19 is the most interesting part of this paper. However, we preferred to structure our manuscript by first comparing COVID-19 and CAP to health (which for CAP has not been reported before), in order to first capture the differences with non-infectious controls, allowing a more logical narrative. In our opinion, this sets the stage for a detailed comparison between the two disease groups, to which we dedicate the final three figures. Therefore we strongly prefer to keep the figures in this order; however, if both the reviewers and the editors feel strongly about changing the order we are of course willing to do so.

6. The number of patients are really few; so comparisons should be deleted from Table 1.

We agree that p-values are not very informative with this sample size, and have therefore removed them in line with the reviewer’s suggestion (see revised Table 1 on page 12).

7. What were the causative pathogens in non-flu CAP?

There was one patient in which H. influenzae was cultured in sputum. This is mentioned in the result section (page 11, line 232) and summarized in the Supplementary File 1. The causative pathogen in the other non-flu CAP patients was not identified, as discussed on page 26, line 560 – 563:

“The causative organism in CAP is typically only identified in approximately 50% of patients. While in this respect our (non-COVID-19) CAP cohort is representative of a general CAP population, patients with CAP-other likely are a more heterogeneous group than patients with a pneumonia caused by one pathogen.”

8. Were PBMCs isolated from critical COVID-19 or just moderate cases? What can be the impact of severity on the three main clusters? Is this clustering associated with outcome and need for dexamethasone treatment?

We agree that disease severity is likely to influence the host response in both patients with CAP and COVID-19. For this reason we intentionally included only patients admitted to the ward and matched the patient groups for disease severity (MEWS score), to make it more likely that differences between groups were due to the disease itself. We therefore do not expect large effects of severity on cell clusters in this cohort, although it would be very interesting to include less/more severely ill patients in future studies (as discussed in comment 4 and the discussion (page 22-23, line 483-486)).

Cell clusters in this study will not be associated with dexamethasone requirements, as this was not standard of care when these patients were included (none of the patients received dexamethasone; this information is provided on page 11, lines 224-226). We considered the group sizes too small and event rates too low (e.g. only 1 patient died) to say anything meaningful about relation between clusters and clinical outcomes.

9. The discussion is too long. The authors need to condense referral to non-COVID-19 as much as possible.

We agree that the discussion featured too much detail that may not add to the main message. In line with this reviewer’s comment, we shortened the discussion significantly (by +- 30%).

10. An integrated view across all the groups with an emphasis of the changes not only in cell specific transcriptional responses, but also at the cell-state level would be of interest.

Indeed, we now included such an overview of cell population differences between the disease groups as main Table 2, which was added in response to point three (Table 2, page 21).

11. The authors are encouraged to provide a complete list of the antibodies used in the CITE seq analysis.

Thank you, this was indeed missing, we added this list to the supplemental information as Supplementary File 2.

12. Technical information would also be of interest including the cell numbers per sample, the median detected gene numbers (especially in view of the 5' sequencing and access to the full transcriptomic response).

Thank you for this good suggestion, we now included this data in the manuscript as Supplementary File 3.

13. A clear limitation is the possibility that some of the major changes between the group reflect the different durations of infection. This may be especially true for the Covid-19 patients where the duration of symptoms was much longer than the other causes of CAP. This should be listed as a limitation of the study.

We agree that the time since infection will influence the immune features of the groups, as patients might be at a different point of their respective immunological trajectories. However, we initially chose this set-up as to ensure that the patients were matched for disease severity, which we discuss in response to point 8. We added this limitation, the difference in time since infection between groups, to the discussion (page 25, line 556 – 558):

“The difference in duration of symptoms prior to hospital admission can be a considered a limitation, as time likely influences the composition of immune cell subsets and transcriptional states.”